# Apoptosis Induced by (−)-Epicatechin in Human Breast Cancer Cells is Mediated by Reactive Oxygen Species

**DOI:** 10.3390/molecules25051020

**Published:** 2020-02-25

**Authors:** Fernando Pereyra-Vergara, Ivonne María Olivares-Corichi, Adriana Guadalupe Perez-Ruiz, Juan Pedro Luna-Arias, José Rubén García-Sánchez

**Affiliations:** 1Departamento de Biología Celular, Centro de Investigación y de Estudios Avanzados del Instituto Politécnico Nacional (Cinvestav-IPN), Ciudad de Mexico C.P. 07360, Mexico; pereyravfer@gmail.com (F.P.-V.);; 2Sección de Estudios de Posgrado e Investigación, Escuela Superior de Medicina del Instituto Politécnico Nacional, Ciudad de Mexico C.P. 11340, Mexico; imoc7@hotmail.com (I.M.O.-C.); adry_quim901@live.com (A.G.P.-R.)

**Keywords:** apoptosis, breast cancer, reactive oxygen species, (−)-epicatechin

## Abstract

(−)-Epicatechin is a phenolic compound with antioxidant activity that is present in natural food and drinks, such as cocoa and red wine. Evidence suggests that (−)-epicatechin exhibits anticancer activity; however, its mechanism of action is poorly understood. Here, we investigated the anticancer effects of (−)-epicatechin and its mechanism of action in breast cancer cells. We assessed the anticancer activity by cell proliferation assays, apoptosis by DNA fragmentation and flow cytometry. The expression of proteins associated with apoptosis was analyzed by the human apoptosis array. MitoSOX^TM^ Red and biomarkers of oxidative damage were used to measure the effect of (−)-epicatechin on mitochondrial reactive oxygen species (ROS) and cellular damage, respectively. (−)-Epicatechin treatment caused a decreasing in the viability of MDA-MB-231 and MCF-7 cells. This cell death was associated with DNA fragmentation and an apoptotic proteomic profile. Further, (−)-epicatechin in MDA-MB-231 cells upregulated death receptor (DR4/DR5), increased the ROS production, and modulated pro-apoptotic proteins. In MCF-7 cells, (−)-epicatechin did not involve death receptor; however, an increase in ROS and the upregulation of pro-apoptotic proteins (Bad and Bax) were observed. These changes were associated with the apoptosis activation through the intrinsic pathway. In conclusion, this study shows that (−)-epicatechin has anticancer activity in breast cancer cells and provides novel insight into the molecular mechanism of (−)-epicatechin to induce apoptosis.

## 1. Introduction

Breast cancer is the most commonly diagnosed malignancy in women of developed countries and is the leading cause of death for women worldwide [1,2]. Two million new cases and a half million breast cancer-related deaths are recorded annually worldwide [3]. Non-modifiable and modifiable risk factors contribute to the development of breast cancer. Indeed, it has been estimated that modifiable risk factors, such as appropriate dietary habits with the consumption of supplements, could account for one-third of all cancers [4,5].

Amongst many promising candidates for dietary supplements are flavonoids, specifically the catechins family, which shows strong anticancer activity [6]. In this context, epidemiological studies in Asian countries have suggested that the low incidence of some cancers is due to the consumption of green tea [7], which is a beverage where the predominant constituents are catechins [8]. Indeed, there is evidence that suggests that the consumption of green tea inhibits the growth of many tumor types [9]. Green tea extract is rich in flavonoids such as (−)-epigallocatechin-3-gallate (EGCG), (−)-epigallocatechin (EGC), epicatechin-3-gallate (ECG), (−)-epicatechin, and (+)-catechin [10]. Previous studies have suggested EGCG as the most potent of all the catechins [11] in its cytostatic properties in many tumor models. However, its poor stability, absorption, and hepatotoxicity have limited its use [12]. Another flavonoid with important biological activity is (−)-epicatechin (Figure 1), which is a molecule that has been described to have antioxidant [13,14], anti-inflammatory [15], cardiovascular [16] and anticancer [17,18,19] properties.

## 2. Results

### 2.1. The Cell Viability of MDA-MB-231 and MCF-7 Cells is Inhibited by (–)-Epicatechin, Dependent on Its Concentration, and Independent of the Seeded Cell Number

To demonstrate the anticancer effect of (−)-epicatechin, two breast cancer cells lines (MDA-MB-231 and MCF-7) were used. MCF-7 and MDA-MB-231 cell lines are considered to be models of the most common and the most aggressive histological subtype of breast cancer, respectively. MCF-7 is the B luminal type, ER and PR (+), whereas MDA-MB-231 is the B basal type and triple-negative: ER, PR, and HER2 (−). Different numbers of MDA-MB-231 and MCF-7 cells were seeded into a 96-well plate and exposed to different concentrations of (−)-epicatechin (see Material and Methods). Then, the cell viability percentage was measured, and the data obtained are shown in Figure 2. As can be observed, the cell viability decreased in a concentration-dependent manner and was independent of the number of cells seeded (Figure 2A,B). To conclusively determine the specificity of the antiproliferative effect observed with (−)-epicatechin, an analysis of these concentrations was performed in noncancerous cells. MCF10A (breast noncancerous cells) and human coronary artery endothelial cells (HCAEC, noncancerous, and nonbreast cells) were exposed to the same (−)-epicatechin concentrations. Figure 2C shows that the flavonoid did not induce any antiproliferative effect in the noncancerous cells, suggesting that the antiproliferative effect of (−)-epicatechin is selective to cancerous cells.

To establish the effectiveness of the flavonoid in the breast cancer cell lines in this study, a logarithmic concentration response curve with 1 × 10^4^ cells was constructed, and the 50% inhibitory concentration (IC_50_) of (−)-epicatechin was calculated (data not shown). Interestingly, in MDA-MB-231 and MCF-7 cells, (−)-epicatechin displayed an IC_50_ of 350 µM.

### 2.2. (−)-Epicatechin Induces Apoptosis in MDA-MB-231 and MCF-7 Breast Cancer Cells

To characterize the antiproliferative effect of (−)-epicatechin over time, we performed a follow-up experiment on the effect for 72 h. MDA-MB-231 and MCF-7 cells were grown in the presence of (−)-epicatechin at its IC_50_ (350 µM) or in the absence of the flavonoid, and its effect at each 24 h interval was examined. As Figure 3A shows, in both cell lines, there was an inhibition in the proliferation at 24 h of incubation with the flavonoid, whereas proliferation had occurred in the untreated control. This antiproliferative effect was maintained over time, with 50% inhibition observed at 72 h. Since this decrease in the cell viability can be explained by an induction of apoptosis, we analyzed whether (−)-epicatechin could be inducing this process. Therefore, we evaluated the cellular DNA cut by endonucleases (a characteristic of apoptosis but not of necrosis) after incubation with the flavonoid. MDA-MB-231 and MCF-7 (1 × 10^6^ cells) were incubated with the IC_50_ value concentration of (−)-epicatechin for 24, 48, and 72 h, after which DNA fragmentation was analyzed by electrophoresis. Interestingly, DNA fragmentation was observed all three times it was analyzed (Figure 3B). These data strongly suggest the induction of apoptosis as the possible mechanism involved in the antiproliferative effect of (−)-epicatechin.

To establish quantitatively the apoptotic effect, MDA-MB-231 and MCF-7 cells were treated with IC_50_ of (−)-epicatechin, which was stained with Annexin V/PI (propidium iodide) and then analyzed by flow cytometry. After treatment, the cells were divided in quadrant 1 to 4 (Q1 to Q4). The cells in Q1 are necrotic cells, which are PI (+). Cells in the Q2 are late apoptotic cells, which are PI (+) and Annexin V (+). Normal cells are in Q3, which are PI (−) or Annexin V (−). In Q4, cells are Annexin V (+) and early apoptotic cells. Figure 3C shows the flow cytometric analysis of apoptosis in MDA-MB-231 and MCF-7, and the results showed that (−)-epicatechin had an apoptotic effect on both cell lines (Figure 3C). The flow cytometry with Annexin V and PI labeling showed that early (43.8%) and late apoptosis (48.5%) were in a similar mode of cell death in MDA-MB-231 cells. Meanwhile, early apoptosis (72.2%) was the predominant mode of cell death in MCF-7 cells (Figure 3C). Interestingly, the treatment of MCF10A cells with (−)-epicatechin did not induce any mode of cell death by apoptosis, corroborating that the apoptotic effect of (−)-epicatechin is selective to cancerous cells.

To explore the molecular basis of the apoptotic effect of (−)-epicatechin in breast cancer cells, the Human Apoptosis Array was utilized to detect the profiling apoptosis proteins generated during treatment with the flavonoid. As shown in Figure 4, in MDA-MB-231 cells treated with (−)-epicatechin’s IC_50_ value concentration, an increase was generated in the expression of the proteins as follows (compare Figure 4A with Figure 4B): TRAIL receptors (R1/DR4, R2/DR5) (C1–C2, C3–C4), pro-caspase 3 (line B9–B10), SMAC/Diablo (line D15–D16), and HTRA2/Omi (lines C21–C22). In addition, the presence of cytochrome C in both conditions was observed (lines B23–24), and an increase in p53 phosphorylation levels (lines D7–D12) was observed only in the presence of (−)-epicatechin. In contrast, (−)-epicatechin in MCF-7 cells generated an increase in the expression of the proteins detected in the control condition (compare Figure 4C with Figure 4D); however, significant differences in the expression of Bad (line B1–2), Bax (line B3–4), FADD (line C5–6), HO-2/HMOX2 (line C13–14), and HSP60 (line C17–18) were observed.

Since this apoptotic profile was related to an antiproliferative effect on breast cancer cells, we corroborated the absence of this apoptotic profile in MCF10A cells (where the flavonoid had no effect). As expected, no apoptotic profile was generated (Figure 4E,F), corroborating the selectivity of the flavonoid on cancerous cells.

### 2.3. (−)-Epicatechin Induces ROS Production and Oxidative Damage in Breast Cancer Cells

As a result of the intracellular accumulation of ROS can induce apoptosis, the effect of (−)-epicatechin on ROS generation was examined by flow cytometer. We evaluated the production of ROS using the fluorescent MitoSOX™ probe. Interestingly, the exposure of MDA-MB-231 and MCF-7 cells to (−)-epicatechin (350 µM) for 72 h induced an intracellular MitoSOX™ fluorescence (Figure 5A,B, respectively). The generation of ROS was observed in both cells, with a higher generation in the MCF-7 cells (Figure 5B). These data show that the induction of apoptosis by (−)-epicatechin in MDA-MB-231 and MCF-7 cells involved an increase in the production of ROS. Due to the high production of ROS generating oxidative damage to biomolecules, the evaluation of this damage was performed in the presence of (−)-epicatechin. Biomarkers of oxidative damage were assessed at 24, 48, and 72 h of incubation with (−)-epicatechin. As shown in Figure 4C, in MDA-MB-231 cells, (−)-epicatechin generated an increase in lipoperoxidation products (malondialdehyde (MDA)), with 72 h as the time with the highest oxidative damage to lipids (Figure 5C). In contrast, oxidative damage to proteins (carbonyl groups) was also observed beginning at 48 h and still observed at 72 h (Figure 5C). When this oxidative damage was analyzed in MCF-7 cells, (−)-epicatechin generated similar damage to lipid and proteins, but this was observed at 72 h of incubation (Figure 5D). These data strongly suggest that the flavonoid had an impact in the mitochondrial compartment, which generated increased ROS production, oxidative damage, and consequently the induction of apoptosis in cancerous cells.

## 3. Discussion

Currently, the use of the chemotherapy in the treatment of breast cancer causes many undesired side effects [21], and for this reason, there is a growing interest in the discovery of new compounds that are safe and more effective in the treatment of this pathology. In this context, natural products have been used for many years as a source for the discovery of these molecules. As previously reported, (−)-epicatechin (a main component of the cacao and a derivative product of cocoa) is a molecule that shows several beneficial health effects such as anti-inflammatory [15], antibacterial [22], cardio protector [23], and cognitive health benefits [24]. However, there are few studies focusing on its anticancer activity and the associated mechanism of action. In this study, we demonstrated that the flavonoid (−)-epicatechin induced an antiproliferative effect in a concentration-dependent manner. Indeed, this proliferative inhibition was selective to breast cancer cells (MCF-7 and MDA-MB-231) and did not affect noncancerous cells (MCF10A and endothelial cells). Interestingly, we found that the antiproliferative effect was followed by DNA fragmentation in the breast cancer cells. These findings are in accordance with previous reports, which have shown the capacity of the flavonoid to induce DNA fragmentation [18]. On the other hand, the quantification of apoptosis by flow cytometry demonstrated the effectiveness of (−)-epicatechin against human estrogen receptor-positive (MCF-7) and receptor-negative (MDA-MB-231) breast cancer cells. Indeed, the occurrence of apoptosis generated by (−)-epicatechin was quantitatively similar in these two cell lines. This finding suggests that (−)-epicatechin can induce the apoptosis of breast cancer cells, irrespective of their receptor status.

Interestingly, the data obtained from the Human Apoptosis Array showed that (−)-epicatechin-activated pathways were related to the induction of apoptosis in cancerous cells. All these data demonstrated that the anticancer effect of (−)-epicatechin can be ascribed to its possible interactions with proteins. Although the beneficial effects from (−)-epicatechin have been explained in relation to its antioxidant capacity, we propose that its anticancer activity and its selectivity could be related to its interaction with a protein in cancerous cells. Thus, we hypothesized that the interaction of (−)-epicatechin with such a protein (possibly a receptor) is initiated at the membrane surface (Figure 6). 

This interaction generates a signaling pathway modulation that results in changes in the redox cellular status. This can be understood as a stimulus that commits cells to increase ROS in the mitochondria and, as a consequence, oxidative damage and the induction of apoptosis.

This proposal is supported by the observations of the increase in the production of superoxide ion (MitoSOX assays) and the increase in the value of the biomarkers of oxidative damage. Furthermore, the evidence that points out ROS generation and an upregulation of dead receptors (DR) as the mechanism of anticancer of other flavonoids opens the possibility that DR can be considered as a possible target of (−)-epicatechin [25]. In this context, it is important to mention that the data obtained in this study clearly demonstrated that (−)-epicatechin upregulated DR (Figure 4B, lane C1–C4), which is an important event in inducing death ligand-induced apoptosis [26,27]. In this study, we demonstrated that ROS generation was important in the apoptosis induced by (−)-epicatechin. We propose that the apoptotic effect in MDA-MB-231 cells results from triggering the extrinsic pathway, increasing ROS, and upregulating the TRAIL receptor (DR4/DR5) (Figure 6). This activation was enhanced through the intrinsic pathway by the opening of the mitochondrial permeability transition (MPT) pore and the leak of pro-apoptotic proteins (cytochrome C, Smac/Diablo, and HtrA2/Omi) into the cytoplasm, activating apoptosis [28] (Figure 6).

In MCF-7 cells, (−)-epicatechin did not generate changes in the TRAIL receptor, although an upregulation of pro-apoptotic proteins such as Bad and Bax (Figure 4D, B1–B2 and B3–B4) was observed. Since the ratio of Bcl-2 and Bad protein is important in apoptosis [29] and Bad protein is upregulated by (−)-epicatechin, we suggest that the flavonoid induced a regulatory signal for Bad, likely via the MAPK and Akt pathways (due to providing well-known regulatory signals for BAD expression) [30,31,32]. This Bad and Bax upregulation induced the leak of cytochrome C, Smac/Diablo, and HtrA2/Omi into the cytoplasm, activating apoptosis through the intrinsic pathway.

The observed differences in the mechanism inductor of apoptosis in our study may be due to the genetic background of MCF7 and MDA-MB 231 cell lines. Considering the differences in their gene expression patterns, these could determine a different action of (−)-epicatechin on MCF-7 and MDA-MB-231 cells, as has been reported for other flavonoids [33]. It was established that flavonoids have beneficial effects on estrogen-driven breast cancer, due its similitude in chemical structure to estrogen [34]. The ability to mimic estrogen could be one of the reasons why (−)-epicatechin displayed a cytotoxic effect in ER-positive MCF-7 cells; however, we observed that (−)-epicatechin showed similar effects on cell viability in MDA-MB-231 cells. For this reason, we propose that (−)-epicatechin could be acted by a different mechanism. Although the effects of (−)-epicatechin on ER-positive and ER-negative breast cancer cells are consistent with observations with other flavonoids [35,36,37], the cellular targets remain unknown.

It is important to mention that the induction of apoptosis in MDA-MB-231 and MCF-7 cells could also be explained considering the expression patterns of the main genes involved in apoptosis such as AKT1 and p53. In MCF-7 cells, an overexpression of AKT1 has been reported (a protein considered antiapoptotic), while in MDA-MB-231 cells, AKT1 shows minor expression. Whether we postulate that (−)-epicatechin induced a downregulation in this protein, this effect on both cells could explain the induction of apoptosis regardless of estrogen status. In line with this, an effect of (−)-epicatechin that increases p53 expression could be related to the induction of apoptosis, due to p53 being a transcriptional factor that activates the expression of genes that inhibit cell proliferation and induce apoptosis [38,39]. In addition, there is evidence that points out that p53 induces apoptosis through mitochondrial disruption [39] and that its upregulation is increased by an oxidative damage, strongly suggesting that the participation of this protein in the apoptotic effect of (−)-epicatechin.

In conclusion, (−)-epicatechin showed an anticancer effect in both ER-positive MCF-7 and triple-negative MDA-MB-231 breast cancer cells. The cytotoxicity resulted from apoptosis through ROS generation and activation of the extrinsic and intrinsic pathways. This study provides evidence of the therapeutic potential of this flavonoid in cancer; however, further experiments are warranted to resolve the precise mechanisms responsible for its anticancer effects.

## 4. Materials and Methods

### 4.1. Reagents

Routine chemicals and (−)-epicatechin were obtained from Sigma Aldrich-México (Toluca, state of México).

### 4.2. Cell Culture and Treatment

MDA-M-B231 and MCF-7 breast cancer cells (American Type Tissue Culture Collection (ATTC), Rockville, MD, USA) were grown in Dulbecco′s Modified Eagle′s Medium (DMEM) media (Life Technologies, Gaithersburg, MD, USA) supplemented with 5% fetal bovine serum (FBS) (BioWest, Miami, FL, USA), 2 mM glutamine, 100 U/mL penicillin, and 100 mg/mL streptomycin in a humidified incubator at 37 °C with 5% CO_2_. MCF-10A cells (nonmalignant breast epithelial cells) were cultured in DMEM/F-12 supplemented with 5% horse serum (Biowest, Miami, FL, USA), 20 ng mL^−1^ epidermal growth factor (Upstate Biotechnology Incorporated, Lake Placid, NY, USA), 10 mg mL^−1^ insulin (Biofluids, Rockville, MD, USA), and 500 ng mL^−1^ hydrocortisone. Primary human coronary artery endothelial cells (HCAECs) were obtained from ATCC (Rockville, MD, USA). HCAECs were grown in Vascular Cell Basal Medium (ATCC PCS-100–030), Endothelial Cell Growth Kit (ATCC PCS-100–040) and supplemented with 10% fetal bovine serum. For the treatment of cells with (−)-epicatechin for the subsequent evaluation of the effects of (−)-epicatechin, MDA-MB231 and MCF-7 were grown in phenol red-free DMEM media (Life Technologies, Gaithersburg, MD, USA) containing 5% FBS. The FBS was charcoal-stripped (to eliminate the oestrogenic effects) only in the MCF-7 cells. Before adding (−)-epicatechin, the cells were cultured for 48 h in phenol-red free DMEM media, after which the (−)-epicatechin was added.

### 4.3. Cell Proliferation Assays

An MTT [3-(4,5-Dimethyl-2-thiazolyl)-2,5-diphenyl-2H-tetrazolium bromide] assay was used to measure MDA-MB-231 and MCF 7 cell growth after (−)-epicatechin treatment. Both cell types (1 × 10^4^ cell/well) were cultured in 96-well plates with phenol red-free DMEM media as outlined above. Then, the (−)-epicatechin was added at different concentrations (50–500 μM), and the cells were cultured for 3 days. The MTT assays were performed by adding 100 μL of MTT (Sigma, St Louis, MO, USA) (3 mg/mL in clear media) to the cells and incubating for 1 h at 37 °C, 5% CO_2_. The media was removed, and 200 μL of DMSO was added to each well to dissolve the formazan crystals. Absorbance at 550 nm was measured with an iMark™ Microplate Absorbance Reader (Bio-Rad Laboratories, Hercules, CA, USA). Each data point was performed in sextuplicate in three different experiments, and the results were reported as the mean absorption ± SD.

### 4.4. Assessment of Apoptosis by DNA Fragmentation

The presence of internucleosomal DNA cleavage in MDA-MB-231 and MCF-7 cells grown in the absence or presence of (−)-epicatechin (1 × 10^6^ cells/60-mm dish) was investigated using DNA gel electrophoresis. Both cell types were harvested from the culture dishes and washed twice in PBS. The cells were lysed in a solution containing 1% SDS, 0.5% Triton X-100, 20 mM EDTA, and 5 mM Tris-HCl, pH 8.0, and incubated overnight with 0.25 mg/mL proteinase K at 37 °C. The DNA was extracted in phenol/chloroform, ethanol precipitated, resuspended in 10 mM Tris-HCl and 1 mM EDTA (pH 8.0), and incubated with deoxyribonuclease-free ribonuclease (80 mg/mL; for 1 h). The DNA (5 μg/lane) was electrophoresed on 2% agarose gels and visualized by ethidium bromide staining.

### 4.5. Quantification of Cell Apoptosis by Annexin VFITC/PI Staining

Cell death mediated by apoptosis was examined by a double-staining method, using an Alexa Fluor^®^ 488 annexin V/propidium iodide (PI) apoptosis detection kit (Invitrogen). Briefly, breast cancer cells (3 × 10^5^) grown in the absence or presence of (−)-epicatechin for 24 h were collected in 12 × 75 mm^2^ polystyrene tubes and centrifuged at 1500× *g* for 5 min at 4 °C. The supernatants were removed by aspiration and the cell pellets were resuspended in 1 mL PBS. Then, the cells were centrifuged as before, and the supernatants were again aspirated. Then, the cells were resuspended in 100 µL of annexin-binding buffer (which was included in the kit) and stained with 5 µL with Fluor^®^ 488 annexin V/1 µL PI dyes for 15 min at room temperature in the dark. The cells were diluted with 400 µL of annexin-binding buffer and then evaluated by flow cytometry (BD FACSAriaTM Fusion). Controls included cells that were unstained or stained with Annexin V or PI, and they were used to calibrate the FACSAria analyzer for each experiment. Early apoptosis was designated as annexin positive/PI negative, and late apoptosis was designated as annexin positive/PI positive, whereas necrosis was defined as annexin negative/PI positive.

### 4.6. Human Apoptosis Antibody Array

The changes generated by (−)-epicatechin in the expression pattern of human apoptosis-related proteins were investigated using the Human Apoptosis Array Kit (R&D Systems, OR, USA) in accordance with the manufacturer’s instructions. Approximately 1 × 10^7^ breast cancer cells untreated and treated with (−)-epicatechin were solubilized in lysis buffer and centrifuged at 14,000× *g* for 5 min. The protein concentrations of the resulting lysates were measured by Lowry’s method [40]. The antibody-coated array membranes were blocked in Array Buffer 1 with a 1-h incubation on a rocking platform shaker. Then, the membranes were incubated with 300 µg of protein from each sample at 4 °C with gentle shaking overnight. The following day, the membranes were washed three times with 1X washing buffer on a rocking platform and then incubated with 1.5 mL reconstituted biotinylated antibodies for 1 h on a rocking platform. Thereafter, the biotin-conjugated antibodies were removed and again washed three times with 1X washing buffer. A 1:2000 dilution of the streptavidin-HRP in 1X Array Buffer was used to incubate each of the membranes for 30 min. Following incubation, the arrays were washed three times with 1X wash buffer. The membrane intensity and pixel densities were acquired using c-Didigit (LICOR, Lincoln, NE, USA). The relative differences in the expression levels of each protein in untreated and treated cells were determined by comparing signal intensities among the array images after subtraction of the background signals.

### 4.7. Measurement of Reactive Oxygen Species (ROS) and Biomarkers of Oxidative Damage

ROS produced by the mitochondria was detected using a MitoSOX^TM^ Red-based flow cytometric assay. MitoSOX^TM^ Red is oxidized by superoxides once the dye enters the mitochondria and exhibits a red fluorescence in the cytoplasm. MDA-MB-231 and MCF-7 (3 × 10^5^) cells were cultured in the absence and presence of (−)-epicatechin (at the IC_50_ value of 350 µM) for 24 h. After that, the media was removed, and the cells were washed with Hank′s Balanced Salt Solution (HBSS, Thermo Fisher Scientific. Grand Island, N.Y., USA). Clear media supplemented with MitoSOX™ Red (2.5 µM) was added to the cells and incubated for 20 min. Then, the cells were washed gently three times with HBSS and harvested with PBS-EDTA (5 mM). The cellular suspension was centrifuged (2500× *g*, 5 min), and the cellular pellet was resuspended in warm PBS for detection. Briefly, the tube was transferred to a BD FACSAria^TM^ Fusion flow cytometer for recording the mean fluorescence intensity and percentage of stained cells.

To evaluate the oxidative damage induced by (−)-epicatechin in breast cancer cells, biomarkers of oxidative damage to lipid and proteins were analyzed before and after treatment with the flavonoid. Briefly, MDA-MB-231 and MCF-7 (1 × 10^7^) cells were treated with 350 µM of (−)-epicatechin (the IC_50_ value) for 24, 48, and 72 h. The media was recovered, and the cells were harvested by adding 3 mL of cold phosphate-buffered solution and then scraping the cells. The media and cells were pooled and centrifuged at 3500 rpm for 10 min. The cells were recovered and resuspended in 210 µL of saline solution for their disruption by homogenization (10 strokes by Brinkmann Polytron PT 10/35 Blade-type). The homogenate was centrifuged at 3500 rpm for 5 min, and the pellet was recovered and resuspended in 210 µL of saline solution. One hundred microliters of this solution was used to determine the values of the lipidic oxidative products by the 1-methyl-2-phenylindole (MPI) method [41]. The other 100 µL was mixed with 2,4-dinitrophenylhydrazine (DNPH) to determine the values of the carbonyl groups according to Dalle–Donne′s method [42]. Finally, 10 µL was used to determine the total protein concentration using the method of Lowry [40].

## Figures and Tables

**Figure 1 molecules-25-01020-f001:**
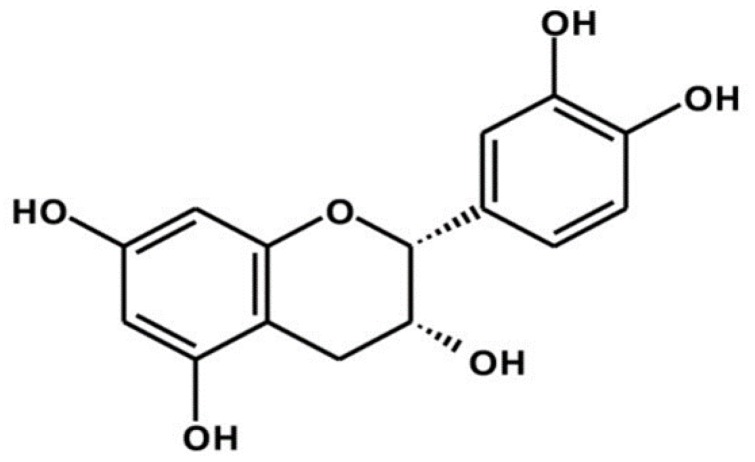
Molecular structure of (−)-epicatechin. The (−)-epicatechin molecule constitutes the most abundant flavonoid in cocoa [20], and, in comparison to (−)-epigallocatechin-3-gallate (EGCG), it has not shown any side effects. Although these features make (−)-epicatechin an interesting molecule for the development of novel phytopharmaceuticals against cancer, little is understood about the mechanism of action involved. In the current study, the anticancer activity of (−)-epicatechin was investigated at the molecular level, focusing on apoptotic cell death in breast cancer cells.

**Figure 2 molecules-25-01020-f002:**
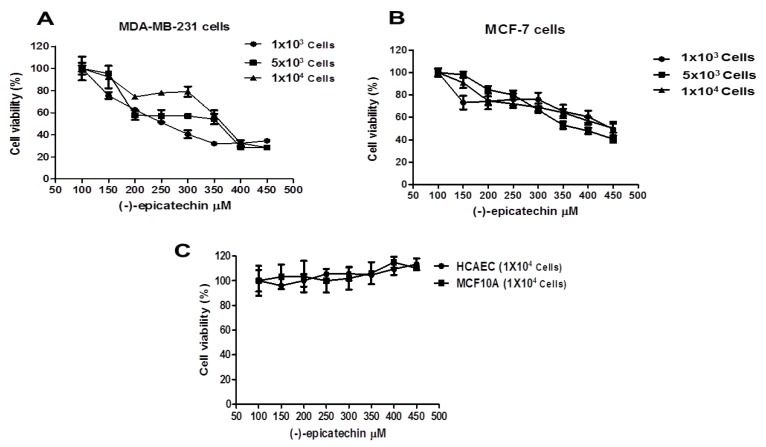
Effects of (−)-epicatechin on cell viability in breast cancer cells. (**A**) MDA-MB-231, (**B**) MCF-7 (1 × 10^3^, 5 × 10^3^ or 1 × 10^4^ cells), and (**C**) human coronary artery endothelial cells (HCAEC) and MCF10A (1 × 10^4^ cells) were grown onto 96-well plates and allowed to grow for 24 h. Then, the cells were treated with (−)-epicatechin (100–450 μM) for 72 h. The number of viable cells was determined by [3-(4,5-Dimethyl-2-thiazolyl)-2,5-diphenyl-2H-tetrazolium bromide] (MTT) assay. Each data point was performed in sextuplicate in three different experiments, and the results were reported as the mean absorption ± SD.

**Figure 3 molecules-25-01020-f003:**
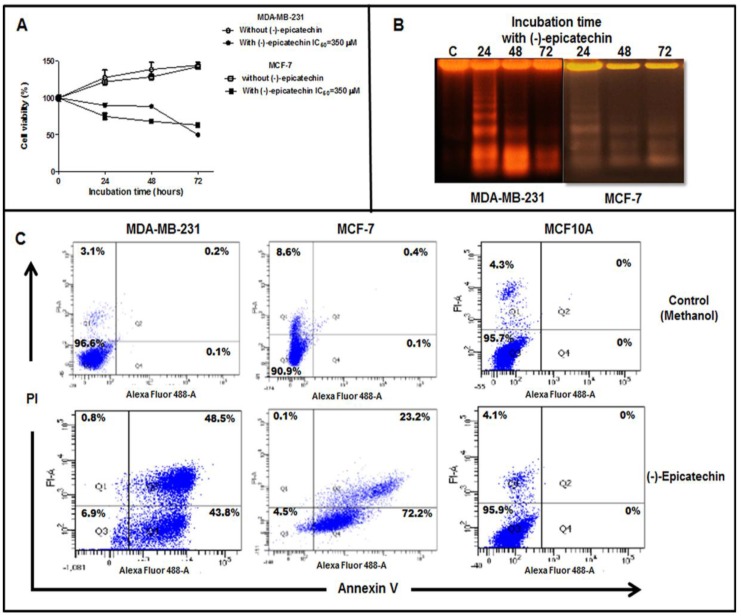
(−)-Epicatechin decreases cell proliferation and induces DNA fragmentation and apoptosis. MDA-MB-231 and MCF-7 cells were grown in the indicated conditions, and the cell viability and DNA fragmentation were assessed at the given time periods. (**A**) Decreasing cell viability in the presence of (−)-epicatechin demonstrated its effects on cell proliferation. (**B**) DNA fragmentation of MDA-MB-231 and MCF-7 cells at the indicated times in the presence of (−)-epicatechin. (**C**) MDA-MB-231, MCF-7, and MCF10A were grown in the absence or presence of (−)-epicatechin and stained with Annexin V/PI (propidium iodide) and subjected to flow cytometric analysis. The four quadrants represent living cells (Annexin V–PI–), early apoptotic (Annexin V+PI–), late apoptotic (Annexin+PI+), or necrotic (Annexin V–PI+) stages. The percentages of cells obtained are indicated in each quadrant, and the results are representative of three similar experiments.

**Figure 4 molecules-25-01020-f004:**
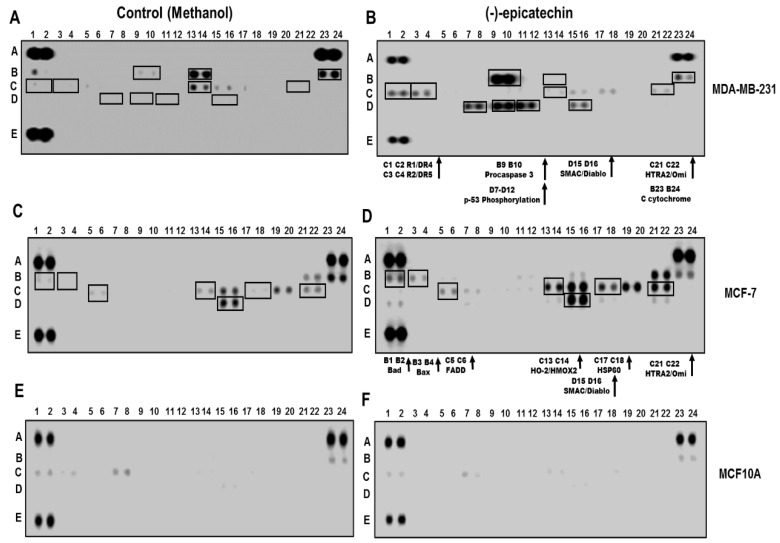
Human Apoptosis Antibody Array analysis in response to (−)-epicatechin. Whole cell lysates from MDA-MB-231, MCF-7, and MCF10A cell lines were prepared from either cells left untreated or treated with (−)-epicatechin. (**A**) and (**B**) MDA-MB-231, (**C**) and (**D**) MCF-7, and (**E**) and (**F**) MCF10A cell lines. Each protein was spotted in duplicate. The pairs of dots in each corner are the positive controls. For the samples exposed to (−)-epicatechin, each pair of the most positive protein dots is denoted by coordinates (letter and number), along with the identity of the corresponding protein.

**Figure 5 molecules-25-01020-f005:**
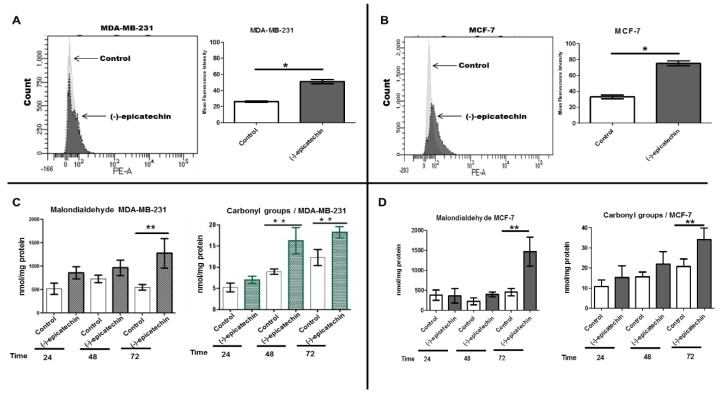
Effects of (−)-epicatechin on mitochondrial reactive oxygen species (ROS) production and oxidative damage. Panels **A** and **B** show the corresponding histograms and mean fluorescence intensity detected in the control cells (without) and the (−)-epicatechin-treated MDA-MB-231 and MCF-7 cells, respectively. The data represent the means ± SD (n = 3). * *p* < 0.05, compared to the control group. **C** and **D** show the values of malondialdehyde (MDA) and carbonyl groups that were generated by (−)-epicatechin on MDA-MB-231 and MCF-7 cells, respectively. The data are expressed as the means ± SD and analyzed by Student’s *t*-test (* *p* < 0.05).

**Figure 6 molecules-25-01020-f006:**
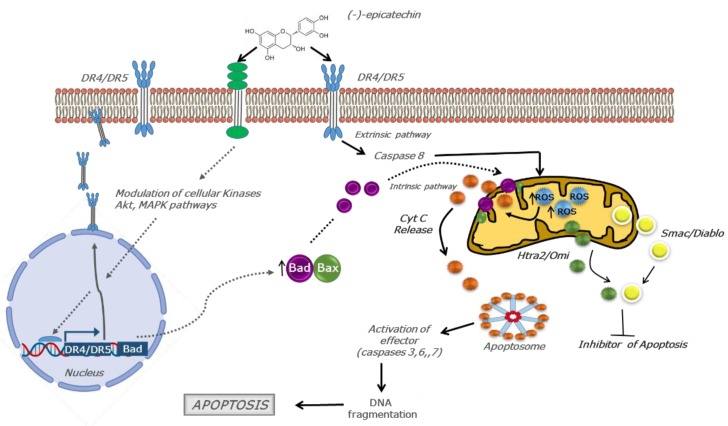
Schematic representation of (−)-epicatechin anticancer activity in MDA-MB-231 and MCF-7 cells. The anticancer effect in MDA-MB-231 cells was possibly triggered through TRAIL receptor interaction (DR4/DR5) and its upregulation. This extrinsic pathway activation was enhanced through the intrinsic pathway, resulting in caspases-dependent apoptosis and enhanced by the modulating of inhibitors of apoptosis by Smac/Diablo and HtrA2/Omi (continuous arrow). In MCF-7 cells, an anticancer effect was evident through the interaction with another receptor, the modulation of cellular kinases, and the upregulation of pro-apoptotic proteins such as Bad and Bax. This Bad and Bax upregulation induced the leak of cytochrome C, Smac/Diablo, and HtrA2/Omi into the cytoplasm, activating apoptosis through the intrinsic pathway. These mechanisms are tightly related to ROS generation (dashes arrow). (Arrows indicate activation, ⊥ indicates inhibition).

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
