# Peer review of "Apoptosis Induced by (−)-Epicatechin in Human Breast Cancer Cells is Mediated by Reactive Oxygen Species"

_molecules, 2020, doi:10.3390/molecules25051020_

Round 1
Reviewer 1 Report
Interesting paper. Quite well designed. Good and complex methodology. The Discussion section looks rather unconventional with an original figure no 6 placed there? The authors don t mention if this is modified or completely original.
Author Response
Dear Reviewer
Thank you for your observations to our manuscript “Apoptosis Induced by (-)-Epicatechin in Human Breast Cancer Cells is Mediated by Reactive Oxygen Species”
I send to you the manuscript with the changes suggested.
Reviewer 1
Comments and Suggestions for Authors
Interesting paper. Quite well designed. Good and complex methodology. The Discussion section looks rather unconventional with an original figure no 6 placed there? The authors don t mention if this is modified or completely original.
R= The figure 6 was placed to end of discussion.
R= The figure is original, and development of data obtained of this investigation.
We thank you in advance for your comments.
Reviewer 2 Report
The work of Pereira-Vergara and cols. analyses the mechanism by which (-)-epicatechin exerts its anticancer effects in a model of in vitro breast cancer. This issue has been assessed in deep by several groups. However, the paper is also original because authors demonstrated the implication of mitochondria, at least in MCF-7 cellsThe manuscript is well written and the state-of-the-art is adequate to the field of research. However, following points should be addressed to strengthen the manuscript to have maximal impact on the readers:
Different apoptotic pathways could be induced by (-)-epicatechin due to the genetic background of MCF7 (ER+) and MDA MB 231 (ER-) cell lines. This should be included in the discussion section. In line with this, authors should include SKBR3 (Her2+) cell line in the study. Apoptosis has been assessed by DNA fragmentation, an obsolete assay. Please, use other techniques such as Annexin/PI by flow cytometry.
Author Response
Dear Reviewer
Thank you for your observations to our manuscript “Apoptosis Induced by (-)-Epicatechin in Human Breast Cancer Cells is Mediated by Reactive Oxygen Species”
I send to you the manuscript with the changes suggested.
Reviewer 2
Comments and Suggestions for Authors
The work of Pereira-Vergara and cols. analyses the mechanism by which (-)-epicatechin exerts its anticancer effects in a model of in vitro breast cancer. This issue has been assessed in deep by several groups. However, the paper is also original because authors demonstrated the implication of mitochondria, at least in MCF-7 cells The manuscript is well written and the state-of-the-art is adequate to the field of research. However, following points should be addressed to strengthen the manuscript to have maximal impact on the readers:
Different apoptotic pathways could be induced by (-)-epicatechin due to the genetic background of MCF7 (ER+) and MDA MB 231 (ER-) cell lines. This should be included in the discussion section.
R= In the discussion was included a paragraph (page6,7 and 8), where the induction of apoptosis by (-)-epicatechin was discussed considering the genetic background of the cell lines studied.
In line with this, authors should include SKBR3 (Her2+) cell line in the study.
R= The inclusion of SKBR3 cells in the study was an interesting observation, however, it was not included due the time. At this point, we needed development several determinations (cell proliferation assays, human Apoptosis Antibody Array analysis, Annexin V, measurement of ROS and biomarkers of oxidative damage) in a short period of time (20 days). Although, we did not include SKBR3 in this study, now we have begun to study the effect of (-)-epicatechin on this cell line. Our apologies in this point, but this was out of our possibilities.
Apoptosis has been assessed by DNA fragmentation, an obsolete assay. Please, use other techniques such as Annexin/PI by flow cytometry.
R= Quantification of apoptosis was performed by flow cytometry; the methodology was included in material and methods (page 9, section 4.5). The data obtained are shown in the figure 3C (page 4).
We thank you in advance for your comments.
Reviewer 3 Report
The article “Apoptosis Induced by (-)-Epicatechin in Human 2 Breast Cancer Cells is Mediated by Reactive Oxygen 3 Species” showed the role of epicatechin on breast cancer cells and studied related mechanism. The experiments are systemically organized and the paper is well written, but the manuscript could be improved with some changes.
Introduction
It is well written from breast cancer epidemiology and current therapies, and explanation about epicatechin.
On Result section, I recommend more details about breast cancer cell types. The reason of choice of two cell types. On the experimental scheme, it is good that include noncancerous cell groups. Eating epicatechin at IC50 level is feasible? I recommend one more apoptosis experiment by Annexin V/PI staining and FACS. If possible, further mechanism study can be performed. For example, ERK/MEK pathway.Author Response
Dear Reviewer
Thank you for your observations to our manuscript “Apoptosis Induced by (-)-Epicatechin in Human Breast Cancer Cells is Mediated by Reactive Oxygen Species”
I send to you the manuscript with the changes suggested.
Reviewer 3
Comments and Suggestions for Authors
The article “Apoptosis Induced by (-)-Epicatechin in Human 2 Breast Cancer Cells is Mediated by Reactive Oxygen 3 Species” showed the role of epicatechin on breast cancer cells and studied related mechanism. The experiments are systemically organized and the paper is well written, but the manuscript could be improved with some changes.
Introduction
It is well written from breast cancer epidemiology and current therapies, and explanation about epicatechin.
On Result section, I recommend more details about breast cancer cell types. The reason of choice of two cell types. R= In the section of results (page 2) a short paragraph was included explaining the reason to use MDA-MB-231 and MCF-7 cells.
On the experimental scheme, it is good that include noncancerous cell groups. Eating epicatechin at IC50 level is feasible? R= In this respect, I would like to comment you that of our knowledge, there are not studies in human on consumption of epicatechin. However, there are an important number of studies on the beneficial effects of the consumption of products with high content of epicatechin as cacao, cocoa derived products (dark chocolate), wine etc. Indeed, all these products have high concentrations of epicatechin that exceed the IC50. On the other hand, do not exist evidences of toxic effects when epicatechin has been used in experimental animals. Finally, I would like to point out, that whether well, there are a lot of evidences of the important effects of epicatechin in the health, little is understood about of its anticancer effect.
I recommend one more apoptosis experiment by Annexin V/PI staining and FACS. If possible, further mechanism study can be performed. For example, ERK/MEK pathway.
R= Quantification of apoptosis was performed by flow cytometry; the methodology was included in material and methods (page 11, section 4.5). The data obtained are shown in the figure 3C. For reason of time (20 days), it was not possible to perform more studies on signaling pathways, our apologies in this point, but this was out of our possibilities.
We thank you in advance for your comments.
Round 2
Reviewer 2 Report
I recommend to publish the article in its current form.